# Kinetics of the Synthesis of Aluminum Boride by the Self-Propagating High-Temperature Synthesis Method

Sestager Khusainovich Aknazarov [1,2], Alibek Zhumabekovich Mutushev [1,2,*], Juan Maria Gonzalez-Leal [2,3], Olga Stepanovna Bairakova [2], Olga Yuryevna Golovchenko [1,2], Natalia Yuryevna Golovchenko [2] and Elena Alexandrovna Ponomareva [2]

1   Department of Chemistry and Chemical Technology, Al-Farabi Kazakh National University, Almaty 050038, Kazakhstan
2   LLC Scientific Production—Technical Center Zhalyn, Almaty 050038, Kazakhstan
3   Department of Condensed Matter Physics, Faculty of Sciences, University of Cadiz, Puerto Real, 11510 Cadiz, Spain
*   Correspondence: alibek_090@mail.ru

**Abstract:** The influence of certain factors on the kinetics of the process of obtaining aluminum borides (burning rate, ingot formation, and phase separation) was investigated. In this study, we report the registration of diboride using the SHS protocol. The synthesis of aluminum diboride from boric anhydride occurred by the aluminothermic method. The initial components were boron trioxide and aluminum in the form of powders. Researchers paid special attention to the degree of grinding of the charge fluxing substances. The influence this had on the rate of development of the degree of charge concentration was studied. To calculate the degree of charge, a composition was chosen according to the speed obtained from a number of experiments where melting was carried out with the following charge densities in g/cm$^3$: 0.80; 1.08; 1.18; 1.74. The method of melting was ignition from above. The experimental results allowed us to conclude that the nature of the change in the combustion rate of the system, where there was an excess of the reducing agent in the charge, is the same. An increase in the combustion rate, where there was an excess of aluminum of up to 20%, was likely due to the fact that the reaction area of the charge components increased. In addition, an increase in speed can be explained by a decrease in heat losses due to a reduction in the melting time. With an increase in excess aluminum above 20% of the stoichiometry, the observed decrease in the combustion rate can be explained by a decrease in the specific heat of the process due to the melting of the excess aluminum, which played the role of a ballast.

**Keywords:** boride; combustion rate; phase separation; mechanoactivation

## 1. Introduction

An important trend in the improvement of one of the most resource-intensive modern industries, metallurgy, is the reduction of the consumption of resources, both material and energy. Along with traditional types of energy (such as electrical), the production of ferroalloys and rare metals uses the chemical energy of reduction of metal oxides by reducing metals with a high affinity for oxygen, such as aluminum, silicon, boron, etc., as well as their alloys. The undoubted advantage of the latter is the unlimited reserves in nature, as well as their environmental safety.

The development of new equipment and technology is largely associated with the creation of new materials capable of operating at high speeds and temperatures. To meet these requirements, materials must have a complex set of physicochemical and mechanical properties. Recently, many studies have been directed towards the creation of oxygen-free refractory compounds based on aluminum borides, which are characterized by high refractory properties and high energy characteristics. Aluminum borides are a promising energy compound for high-energy systems for various purposes [1,2].

Aluminum borides are introduced into steels and some alloys of non-ferrous metals (aluminum, copper, nickel, etc.), in order to make them more fine-grained and significantly improve their mechanical properties. The addition of boron to high-speed steel significantly improves its cutting properties. This is explained by the fact that metal borides that are formed at high temperature have high hardness and wear resistance [1,3].

Composite material aluminum-boron is characterized by a combination of high strength, endurance limit, and by an elastic modulus with high fracture work Aluminum borides are used in mechanical engineering for the production of wear-resistant parts subject to high loads and aggressive environments [4].

Due to their low density, chemical resistance, strength, and other valuable properties, aluminum borides are also used in nuclear power engineering.

Currently, there is no effective technology for its production. Furnace technologies, using the method of metallothermy, are the most widely used [2]. All over the world, methods are being developed for the production of metal borides in different directions and in various ways.

One of the methods is the technology of self-propagating high-temperature synthesis (SHS), in which the process of obtaining the target materials is carried out using the heat of exothermic reactions of reactants; this is an alternative to reducing the energy consumption of metallurgical processes.

The SHS method used [5,6] is rather new, but it has become widespread in the field of obtaining refractory and various composite materials. The method is easy to implement, environmentally friendly, and does not require high material costs or complex technological equipment. The product obtained by this method is of high purity. This technology makes it possible to obtain materials with a given composition. Additionally, by-products do not require special disposal, as they do not contain harmful impurities, and can be used in the production of new commercial materials (abrasives, cements, building fillers, etc.).

## 2. Materials and Methods

To determine the possibility of the reaction of reduction of boric anhydride with aluminum according to Equation (1), Gibbs energy was calculated depending on the temperature, ranging from 298 to 1900 K. The value of $\Delta G$ was determined to be in the range of $-548.79$ to $-273.65$ kJ/mol, which indicated that the reaction proceeded in a spontaneous mode. Below 298 K, the process does not occur spontaneously [7].

The synthesis of aluminum boride was carried out by the aluminothermic method in the SHS mode. The starting components were powders. To determine the burning rate of the charge to produce aluminum boride, a laboratory setup was assembled, shown in Figure 1. Holes were drilled transversely in the lateral surface of the crucible to place two thermocouples 5 cm apart. The ignition was carried out from above by an electric pulse applied to the nichrome spiral through the adjustable laboratory autotransformer. Chromel-alumel thermocouples were used to fix the burning rate. The signal from the thermocouple was recorded on a two-channel USB oscilloscope (Acute Technology, model TS2212F). The linear rate of synthesis was determined by the calculating the ratio of the distance traveled by the combustion front between the thermocouples, and the time.

Since boric anhydride is a hard-to-reduce oxide with a high affinity for oxygen, when conducting a reduction with aluminum while ensuring the completeness of the reaction, melting of alumina slag, good phase separation of the alloy and slag, it is necessary to increase the thermal load. For the out-of-furnace process, food additives are usually used; these are compounds that easily split off their oxygen and release a large amount of heat, compensating for the oxygen that is missing, and have significant properties in terms of the parameters of metallothermic processes and the degree of metal extraction [8,9].

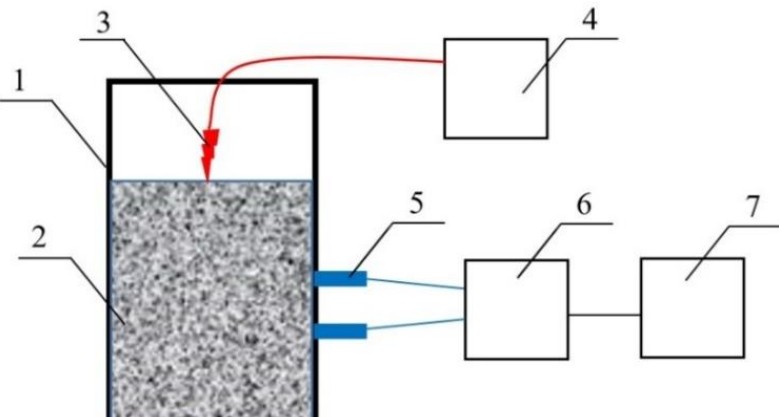

**Figure 1.** Diagram of the installation for determining the combustion rate of the charge: (1) crucible; (2) charge; (3) nichrome spiral; (4) adjustable laboratory autotransformer; (5) thermocouples; (6) dual-channel USB oscilloscope; (7) personal computer.

The prepared mixture, consisting of a stoichiometric ratio of boric anhydride and aluminum according to Equation (1), was loaded into the crucible and compacted in portions as it was loaded. The interaction did not occur either during local initiation or at the charge heating temperature up to 700 °C. A sinter was formed.

The combustion temperature was determined based on of thermal electromotive force, according to GOST R 50431-92-Thermocouple, and was over 1900 °C. The linear burning rate was 0.88 mm/s, and the mass burning rate was 2.5 g/s. The yield of the alloy from the calculated value was 76.8%. The content of aluminum boride in the alloy was 92.5%.

## 3. Results

To determine the possibility of the reaction of reduction of boric anhydride with aluminum according to equation,

$$B_2O_3 + 3\,Al = AlB_2 + Al_2O_3 \tag{1}$$

the Gibbs energy was calculated depending on the temperature in the studied range. The negative value of $\Delta G = -577.144$ kJ/mol indicates that the reaction proceeds in a spontaneous mode [5].

Since boric anhydride is a hard-to-reduce oxide with a high affinity for oxygen, when conducting a reduction with aluminum while ensuring the completeness of the reaction, melting of alumina slag, good phase separation of the alloy and slag, it is necessary to increase the thermal load. For the out-of-furnace process, heating additives are usually used; these are compounds that easily split off their oxygen and release a large amount of heat, compensating for the oxygen that is missing. They significantly affect the parameters of metallothermic processes and the degree of metal extraction [5,6].

We studied what effect different amounts of heating additive (specifically potassium nitrate) had on the rate of the process. The amount of heating additive varied.

As shown in Table 1, an increase in the amount of nitrate led to an acceleration of the burning rate. The development of the combustion process occurred with the addition of 15–20% of saltpeter from the total volume of the charge. Table 1 presents the variations of the conducted experiments.

**Table 1.** Effect of the amount of heating additive on the combustion rate of the charge.

| № | Components | Content, g | Content, % | Burning Rate, g/s | Note |
|---|---|---|---|---|---|
| 1 | $B_2O_3$<br>Al | 50.00<br>58.02 | 46.3<br>53.7 | – | Not initiated, furnace combustion |
| 2 | $B_2O_3$<br>Al<br>$KNO_3$ | 50.00<br>60.39<br>5.00 | 43.3<br>52.3<br>4.3 | – | Not initiated, furnace combustion |
| 3 | $B_2O_3$<br>Al<br>$KNO_3$ | 50.00<br>62.02<br>10.00 | 41.0<br>50.8<br>8.2 | – | Furnace combustion, slow |
| 4 | $B_2O_3$<br>Al<br>$KNO_3$ | 50.00<br>66.90<br>20.00 | 36.5<br>48.9<br>14.6 | 0.54 | Ignition after secondary ignition |
| 5 | $B_2O_3$<br>Al<br>$KNO_3$ | 50.00<br>69.19<br>25.00 | 34.7<br>48.0<br>17.3 | 0.68 | The spin nature of combustion |
| 6 | $B_2O_3$<br>Al<br>$KNO_3$ | 50.00<br>71.39<br>30.00 | 33.0<br>47.2<br>19.8 | 0.72 | No phase separation |
| 7 | $B_2O_3$<br>Al<br>$KNO_3$ | 50.00<br>83.61<br>35.00 | 29.7<br>49.6<br>20.8 | 0.84 | No phase separation |
| 8 | $B_2O_3$<br>Al<br>$KNO_3$ | 50.00<br>85.84<br>40.00 | 28.4<br>48.8<br>22.7 | 0.97 | No phase separation |
| 9 | $B_2O_3$<br>Al<br>$KNO_3$ | 50.00<br>90.30<br>50.00 | 26.3<br>47.5<br>26.3 | 1.02 | Combustion equal, molten metal |

One of the criteria affecting both the rate of the oxide reduction process, and the percentage of recovery of the recovered metal, is the fineness of the initial components. In the production of metals and alloys by the out-of-furnace aluminothermic method of difficult-to-recover elements, such as Ti, Si, Cr, Zr, B, V, etc., the particle size of oxides should not exceed 0.5 mm. The size of the aluminum powder was selected based on the size of the oxides and the conditions of the process [9,10].

To maximize the development of reduction reactions, the size of the reducing agent was chosen in such a way that after mixing the charge materials in each elementary part of the charge that enters the reaction, the components were in a stoichiometric ratio (taking into account the reducibility coefficient of oxides). The ratio between the size of the oxides and the reducing agent was determined both by the conditions that ensure the most uniform mixing of the charge (which is feasible when using particles of the same size), and by the ratio of their gram-equivalent volumes. For most oxides that are important for aluminothermic processes, the gram-equivalent volume should exceed the gram-equivalent of aluminum by 1.5–2 times [11]. Therefore, the most complete course of aluminothermal reduction can be expected when using oxides and a reducing agent that are close in size.

An increase in the fineness of the reducing agent leads to a decrease in the rate of the combustion process. However, excessive grinding of charge materials can adversely affect the process performance, since the fineness of the reducing agent affects not only the process rate, but also the alloy yield [8]. The regrinding of the reducing agent leads to a decrease in the yield of the target component, since the formed small pellets of the melt "get entangled" in the slag [10].

In practice, the optimal fineness of the reducing agent is usually determined experimentally. In the present study, aluminum grade AP with an activity of 97.8% was used. Before carrying out experiments to determine the effect of particle size distribution, all initial powders were dried. Following this, the powders were sieved into fractions. After the powders were dosed in accordance with the calculated composition, they were thoroughly mixed to evenly distribute the components.

Table 2 shows the dependence of the charge combustion rate on aluminum dispersion.

**Table 2.** The dependence of the charge combustion rate on aluminum dispersion.

| The Size of Aluminum (mm) | The Time of the Process (s) | Combustion Burning Rate (g/cm$^3$ min) |
| --- | --- | --- |
| 0.056 | 23 | 2.04 |
| 0.125 | 35 | 1.35 |
| 0.500 | 28–30 | 1.30 |
| 0.630 | 38 | 1.23 |
| Not distracted | 35–36 | 1.34 |

With aluminum powder particles of 0.63 mm, the lowest burning rate was observed; the dispersal of alloy pellets throughout the slag volume demonstrated poor phase separation. With an aluminum powder size of 0.056 mm or less, the mixture burned violently with a scattering of reaction products along the reactor walls. An alloy ingot was formed, but with large losses. The optimal size for aluminum powder particles of 0.63 mm for a given charge composition was 0.125–0.50 mm, which provided a combustion rate that was sufficient for the completeness of the process, and ensured good phase separation of its products.

An important influence on the rate of the process in any method of melting is the degree of charge compaction [11]. A set of experiments was carried out to determine the effect that the charge density of the selected composition had on the rate of the process. Meltings were carried out with the following charge densities in g/cm$^3$: 0.80; 1.08; 1.18; 1.74. The method of melting was ignition from above. Figure 2 shows the dependence of the burning rate on density.

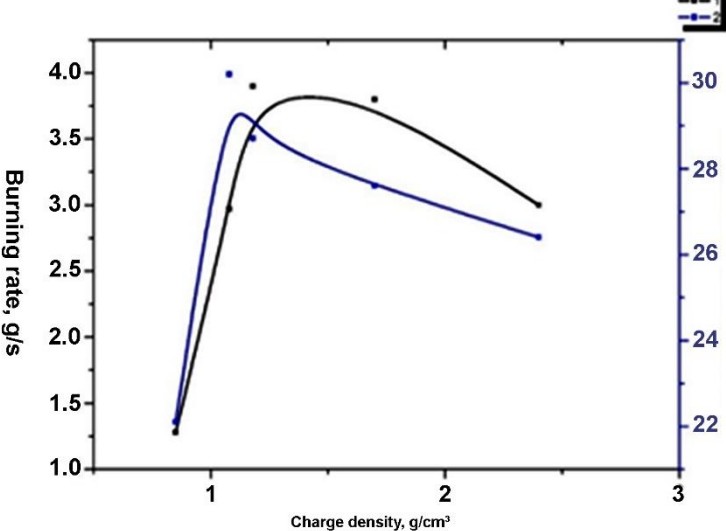

**Figure 2.** The effect of charge density on combustion and boron content in the alloy: (1) burning rate; (2) boron content.

In the experiments performed, the optimal charge density was 1.08–1.18 g/cm$^3$. At these densities, a sufficiently high burning rate developed, and a high boron content was provided in the alloy pellet.

When the charge density increased above 1.18 g/cm$^3$, the rate decreased, and the boron content in the alloy also decreased. The likely reason for the decrease in the speed of the compacted charge is the deterioration of the conditions for the infiltration of the liquid melt into the depths of the charge.

The rate of the synthesis of aluminum boride in the SHS mode, in addition to the dispersity of the initial components and the charge density, is affected by the amount of aluminum. To select the optimal amount of aluminum, a series of experiments was carried out. The stoichiometric amount of aluminum was calculated from the reduction reactions of boric anhydride (1) and potassium nitrate:

$$6 \, KNO_3 + 10 \, Al = 5 \, Al_2O_3 + 3 \, K_2O + 3 \, N_2 \qquad (2)$$

Table 3 shows the options for the composition of the charge and its burning rate which depended on the amount of aluminum. In all experiments, the fineness of aluminum was 0.10–0.25 mm, and the charge density was 1.08–1.18 g/cm$^3$.

**Table 3.** Dependence of the burning rate of the system on the amount of aluminum in the charge.

| № | Components, g | | | Excess Al (%) | Charge Weight (g) | Burning Rate (g/s) | Note |
|---|---|---|---|---|---|---|---|
| | B$_2$O$_3$ | KNO$_3$ | Al | | | | |
| 1 | 50.00 | 25.00 | 68.04 | – | 143.04 | 1.43 | No phase separation, molten metal |
| 2 | 50.00 | 25.00 | 74.84 | 10 | 149.84 | 1.56 | No phase separation, molten metal |
| 3 | 50.00 | 25.00 | 78.25 | 15 | 153.25 | 1.82 | No phase separation, molten metal |
| 4 | 50.00 | 25.00 | 81.65 | 20 | 156.65 | 1.96 | No phase Separation, molten metal |
| 5 | 50.00 | 25.00 | 85.05 | 25 | 160.05 | 1.86 | No phase separation, molten metal |
| 6 | 50.00 | 25.00 | 88.45 | 30 | 163.45 | 1.70 | No phase separation, molten metal |

Based on the data presented, it can be concluded that the combustion rate of the system under study increased with an excess of aluminum content up to 20%. A further increase in aluminum led to a decrease in the burning rate. This is explained by a decrease in the specific heat of the process due to the melting of the excess aluminum, which played the role of a ballast.

The density of some aluminothermic alloys with elements such as barium, boron, or titanium may be close to the density of the resulting slag, which led to a disruption of the normal process of ingot formation [12].

The melting of the charge without fluxing additives did not reflect the typical kinetics (reaction rate, drop rate of the reduced metal, ingot formation, etc.) of the aluminum oxide reduction process.

The influence of fluxing additives on the penetration rate of aluminothermic charges is determined by their technological role. The main purpose of introducing fluxes into the composition of the charge is to obtain slag melts with certain physicochemical character-istics that provide a more complete separation of the metal and slag phases. The main fluxing materials are oxides of calcium, magnesium, and fluorspar. They impact the speed of the process differently.

With the introduction of calcium and magnesium oxides into the composition of the charge, no sharp intensification of the onset of the reaction was observed. The solid calcium oxide used in the charge, the melting point of which exceeds the process tempera-

ture, passed into the melt due to dissolution in liquid high-alumina slag. Due to melting caused by the heat of exothermic reactions, calcium and magnesium oxides reduced the temperature of the melt and, therefore, the rate of melting of the charge as a whole. The decreases in the rate of charge penetration due to the addition of these oxides were established in works [12,13] on the smelting of ferroalloys of titanium, vanadium, niobium, and other alloys.

Fluorspar additives had a different effect on the charge penetration rate. The activation of aluminothermic reduction by fluorspar can be explained by its ability to dissolve alumina due to a decrease in the liquidus temperature of the melt, compared to the melting point of pure aluminum oxide [13]. This, firstly, promotes the removal of a hard oxide film from the surface of the reducing agent particles. Secondly, this ensures the dissolution of alumina. As a result, the initial temperature of the interaction between liquid melts decreases, and the diffusion of reagents to their interface is facilitated, which explains the increase in both the penetration rate and the metal yield.

To study the effect of flux on the penetration rate and yield of the alloy, experiments were carried out with a charge containing calcium oxide and calcium fluoride. The amount of flux varied from 10 to 50% of the amount of aluminum. In all experiments, the content of the main components remained constant; only the content of flux additives changed. All components of the charge had the same particle size distribution. The calcium oxide was fired before use. Table 4 shows the initial data and results of the research.

**Table 4.** Influence of fluxes on the combustion rate.

| № | Components (g) | | | | | Burning Rate (g/s) | Note |
|---|---|---|---|---|---|---|---|
| | $B_2O_3$ | $KNO_3$ | Al | CaO | $CaF_2$ | | |
| 1 | 50.00 | 25.00 | 68.40 | – | – | 1.40 | No phase separation |
| 2 | 50.00 | 25.00 | 68.40 | 5.00 | – | 1.00 | Sintered product molten metal |
| 3 | 50.00 | 25.00 | 68.40 | 7.50 | – | 0.80 | molten metal in the slag phase |
| 4 | 50.00 | 25.00 | 68.40 | 10.00 | – | 0.50 | Poorly initiated, no melt |
| 5 | 50.00 | 25.00 | 68.40 | 15.00 | – | 0.50 | Smoldering combustion |
| 6 | 50.00 | 25.00 | 68.40 | 20.00 | – | – | Not initiated |
| 7 | 50.00 | 25.00 | 68.40 | – | – | 1.43 | No phase separation |
| 8 | 50.00 | 25.00 | 68.40 | – | 3.42 | 1.56 | Heterogeneous melt |
| 9 | 50.00 | 25.00 | 68.40 | – | 6.75 | 1.82 | Heterogeneous melt |
| 10 | 50.00 | 25.00 | 68.40 | – | 6.82 | 1.96 | Speed increase, slag is dense, there is phase separation |
| 11 | 50.00 | 25.00 | 68.40 | – | 8.21 | 2.08 | |
| 12 | 50.00 | 25.00 | 68.40 | – | 10.26 | 2.30 | Good phase separation, ingot formed |
| 13 | 50.00 | 25.00 | 68.40 | – | 12.31 | 2.40 | |
| 14 | 50.00 | 25.00 | 68.40 | – | 13.68 | 2.10 | There are many molten metal in the slag |

Figure 3 shows a dependence graph of the combustion rate of a system based on boron oxide and aluminum on the amount of flux additives.

The reason for the decrease in the combustion rate when CaO is added as a flux is the decrease in the process temperature, which worsens the penetration of the charge due to heat loss from the melting of calcium oxide [13].

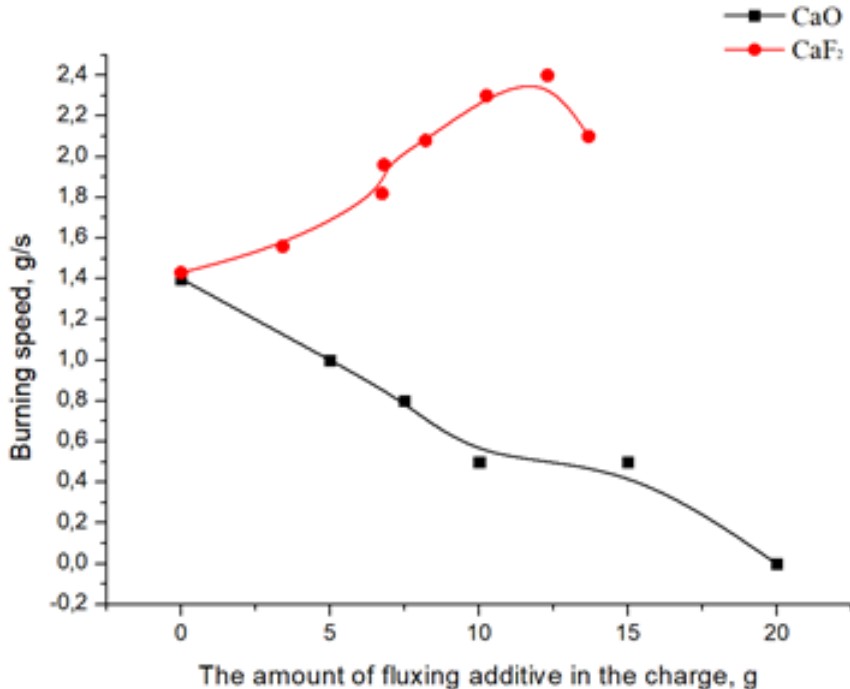

**Figure 3.** Dependence of the burning rate of the charge based on the $B_2O_3$-Al system on the amount of flux.

In addition to the prescription method of influencing the kinetics of the process, there are other methods for its intensification, one of which is mechanoactivation [14,15]. Thus, mechanical activation was used with a combination of the main components of the charge: boric anhydride and aluminum according to the composition, in %: $B_2O_3$ = 24.2, Al = 43.6; $KNO_3$ = 24.2; $CaF_2$ = 8.0. Activation was carried out on a Pulverisette-5 mill with a chamber volume of 250 $cm^2$, a rotation speed of 600 rpm, and an acceleration of 40 g. Balls of various diameters from 5 to 15 mm were used in the ratios, in %: 5 mm, 50; 10 mm, 30; 15 mm, 20, all with a ratio of weights where W:T = 5:1. Activation time was between 2 and 3 min.

After activation, a thermal analysis of the mechanically activated mixture of boric anhydride and aluminum was carried out by using a differential scanning calorimeter (NETZSCH, model STA 449F3). Thermal analysis was carried out to determine the chemical reactions and physical transformations at a constant volume of heating of the powder mixture. Measurement conditions were as follows: sample weight = 12–15 mg, Tmax = 1000 °C, heating rate = 30/20 °C/min. As the results of the analysis showed, at a temperature of about 150 °C, sorption water was removed from all samples, and for sample (c) it began much earlier at 146 °C. At the same time, a decrease in mass was observed on the TG curve (see Figure 4).

Further heating led to melting and subsequent oxidation of aluminum, which was confirmed by the presence of an exothermic effect on the DTA curve, and an increase in mass on TG at temperatures above 600 °C. Moreover, the oxidation of a sample subjected to mechanical activation for 3 min began at a lower temperature of 656.3 °C with a maximum weight gain of 2.40%.

A series of experiments was then carried out to determine the effect of mechanical activation on the combustion rate of the charge of the selected composition, and the yield of the alloy. The data are shown in Table 5.

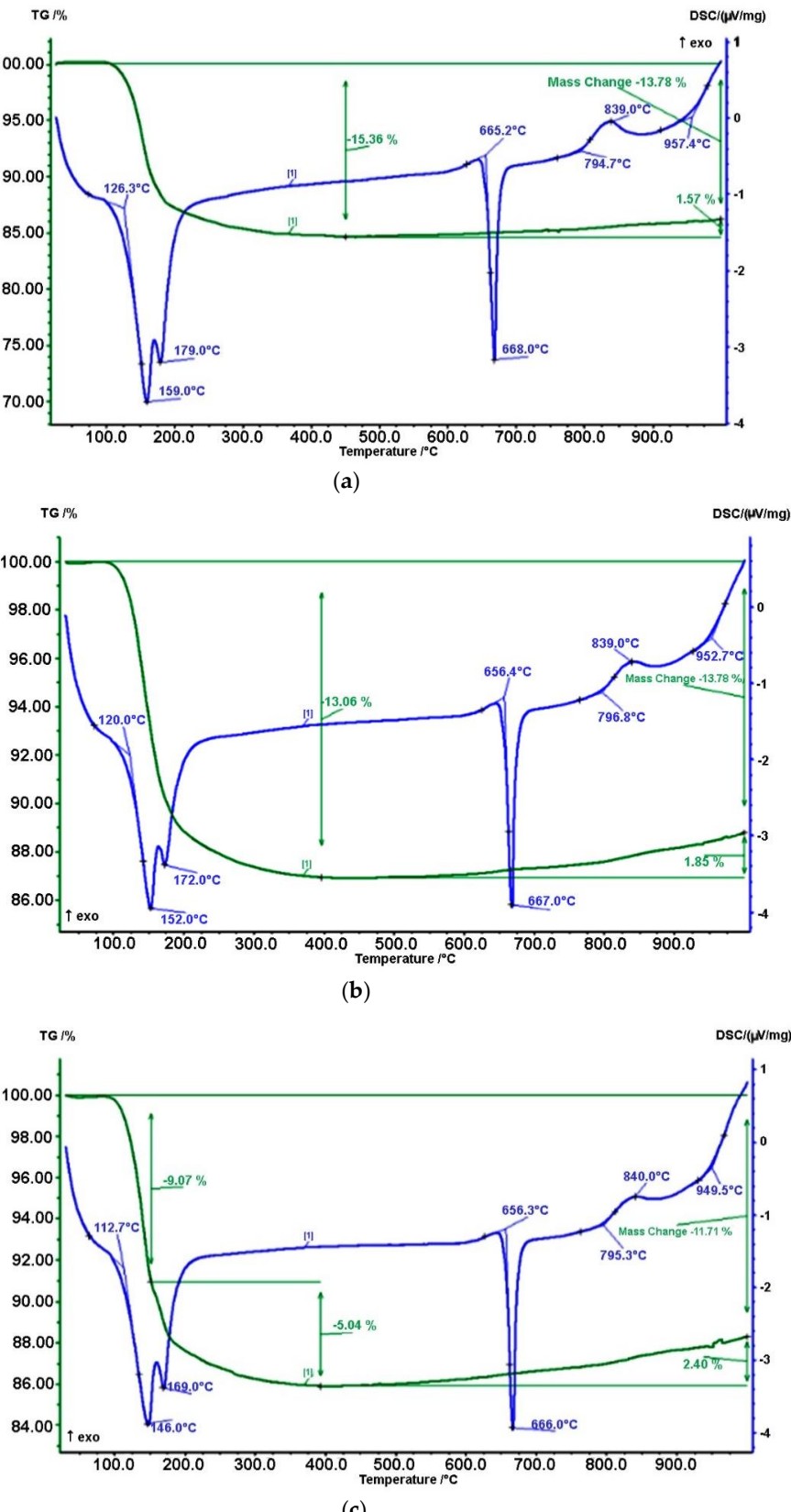

**Figure 4.** Results of thermal analysis of a mixture of boron hydride and aluminum without mechanoactivation (**a**), with mechanoactivation for 2 min (**b**) and with mechanoactivation for 3 min (**c**).

**Table 5.** Dependence of the combustion parameters of the charge on the activation time.

| № | Charge Weight (g) | Activation Time (min) | Linear Combustion Velocity (mm/s) | Mass Rate of Combustion (g/s) | Combustion Temperature (°C) | The Output of the Alloy from the Calculated (%) |
|---|---|---|---|---|---|---|
| 1 | 135.20 | 0 | 1.00 | 3.60 | Over 2500 | 75.80 |
| 2 | 91.10 | 2 | 0.88 | 2.50 | 1900 | 76.00 |
| 3 | 91.10 | 3 | 1.03 | 3.00 | 1900 | 70.58 |

As can be seen from the data obtained, the MA of the main components for 2 min led to a decrease in the combustion rate of the charge, both mass and linear, as well as to a decrease in the process temperature. Additionally, it led to an increase in the yield of the alloy and the formation of a solid bead, which is possibly due to the formation of low-melting eutectics, which contribute to a decrease in the viscosity of the slag component of the melt.

After activation of the mixture for 3 min, a decrease in the ignition temperature, an increase in the burning rate, and a decrease in the yield of the alloy were observed. This can probably be explained by the refinement of the reducing agent, which led to a decrease in the yield of the target component as the resulting small melt pellets do not have time to fully form into a single ingot [15].

Thus, it has been experimentally shown that the 2 min mechanical activation of the main components of the charge can be a technological step in the HF synthesis that increases the yield of the target component.

To create the optimal temperature and speed parameters of the synthesis, using laboratory data on the influence of various factors on the course of the reduction of boric anhydride, the composition of the charge contained boric anhydride, aluminum, potassium nitrate, and fluorspar.

The calculation for the composition was carried out according to the generally accepted method. The amount of aluminum was calculated taking into account the composition and degree of reducibility of all components present in the charge, as well as taking into account the activity of the reducing agent. When calculating the composition, the degree of transition of the reduced elements into the alloy was considered. The degree of reducibility of boron oxide is 65–70% [12]. Table 6 presents the calculated data on the amount of aluminum.

**Table 6.** Calculation of the amount of aluminum with an activity of 97.8%.

| Reaction Equation | Mass of the Recovered Charge Component (g) | Weight of Aluminum According to Stoichiometry (g) | Calculated Weight of Aluminum (g) |
|---|---|---|---|
| $B_2O_3 + 3Al = AlB_2 + Al_2O_3$ | 100.0 | 116.4 | 119.0 |
| $6KNO_3 + 10Al = 3K_2O + 3N_2 + 5Al_2O_3$ | 100.0 | 44.5 | 45.5 |
| TOTAL | | 160.9 | 164.5 |

Based on the data obtained, the amount of fluxing additive in the form of fluorspar was calculated, which was 8% of the aluminum introduced into the charge (45.5 g). Thus, the component composition of the charge with fluxing and heating additives was chosen, in %: $B_2O_3$ = 24.2; Al = 43.64; $KNO_3$ = 24.24; $CaF_2$ = 8.0. The charge of this composition was smelted. Mechanical activation was subjected to a combination of the main components of the mixture for 2 min. The prepared mixture was loaded into the crucible of the installation in Figure 1.

The temperature and burning rate were measured using an A-2 type tungsten-rhenium thermocouple and a two-channel oscilloscope (Acute Technology, model TS2212F).

The combustion temperature was determined on the basis of thermal electromotive force, according to GOST R 50431-92-Termocouplers, which was found to be over 1900 °C. The linear burning rate was found to be 0.88 mm/s, and the mass burning rate was found to be 2.5 g/s. The resulting alloy sample after melting was well separated from the slag component. The yield from the calculated value was 76.8%. The alloy was analysed on an X-ray fluorescence spectrometer (Hitachi, model X-Supreme 8000). According to the results, the aluminium boride phase was 92.5%.

## 4. Conclusions

The influence of various factors on the kinetic characteristics of the synthesis of aluminum borides was studied. Concerning the granulometric composition of the components, and the degree of charge compaction, it was found that the optimal fineness of the reducing agent (Al) was 0.125–0.50 mm, and the optimal fineness of the oxidizing agent ($B_2O_3$) was 0.63–1.25 mm. The optimal charge density was 1.08–1.18 g/cm$^3$, which ensured that the burning rate of the charge, 1.37–2.00 g/cm$^3$·min, was sufficient for the completeness of the process and ensured good phase separation of products with a high boron content in the alloy pellet.

It was also found that an excess of aluminum from the calculated value of 10–20% contributed to the maximum development of the sheet penetration rate. As well, the use of fluoride salts, in particular $CaF_2$, in an amount of 6.0 to 11.0% contributed to an increase in the burning rate of the charge, as well as the formation of mobile liquid slags that improve phase separation at 6.7 to 8.0%; here, the maximum yield of the alloy occurred. It was established that 15.0–20.0% of saltpeter from the total volume of the charge led to an acceleration of the burning rate.

Preliminary preparation of raw materials by mechanical activation of the main components for 2 min led to a decrease in the combustion rate of the mixture, a decrease in the mass combustion rate and linear combustion rate, as well as to a decrease in the process temperature. At the same time, there was an increase in alloy yield and the formation of a solid bead.

The composition of the charge was selected and calculated taking into account the influence of various factors on the course of the reduction of boric anhydride, the degree of reducibility of all components, and the activity of the reducing agent, in %: boric anhydride, 24.2; aluminum, 46.3; potassium nitrate, 24.2; fluorspar, 8.0.

The combustion temperature was determined based on the thermal electromotive force, according to GOST R 50431-92-Thermocouplers, and was over 1900 °C. The linear burning rate was 0.88 mm/s, and the mass burning rate was 2.5 g/s. The yield of the alloy from the calculated value was 76.8%. The content of aluminum boride in the alloy was 92.5%.

**Author Contributions:** Conceptualization, S.K.A. and O.S.B.; methodology, O.Y.G. and J.M.G.-L.; software, N.Y.G.; validation, S.K.A., O.S.B. and A.Z.M.; formal analysis, A.Z.M.; investigation, O.S.B., A.Z.M. and E.A.P.; resources, S.K.A.; data curation, O.S.B. and A.Z.M.; writing—original draft preparation, A.Z.M.; writing—review and editing, J.M.G.-L. and A.Z.M.; visualization, O.S.B., E.A.P. and N.Y.G.; supervision, S.K.A.; project administration, S.K.A.; funding acquisition, S.K.A. All authors have read and agreed to the published version of the manuscript.

**Funding:** This research was funded by the Committee of Science of the Ministry of Education and Science of the Republic of Kazakhstan [Grant number: AP08857190 (2020–2022)].

**Institutional Review Board Statement:** Not applicable.

**Informed Consent Statement:** Not applicable.

**Data Availability Statement:** Not applicable.

**Conflicts of Interest:** The authors declare no conflict of interest.

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
