# Peer review of "Kinetics of the Synthesis of Aluminum Boride by the Self-Propagating High-Temperature Synthesis Method"

_ceramics, doi:10.3390/ceramics5030033_

Round 1
Reviewer 1 Report
1. Suggest not to use SHS in the title, using the full name instead.
2. Why the burning rate is different in blend with CaO and CaF2. And there is a peak point in the CaO sample, can you explain why?
3. The figures should be better represented, like the Figure 5. In three separate figures, it is hard for readers to compare.
4. The legend in Fig 8 is too small, hard to read.
Author Response
Response to Reviewer 1 :
Point 1.Suggest not to use SHS in the title, using the full name instead.
Response 1. Reviewer suggestion has been addressed in the revised version of the manuscript.
Point 2. Why the burning rate is different in blend with CaO and CaF2. And there is a peak point in the CaO sample, can you explain why?
Response 2. The reason for the decrease in the burning rate when CaO is added as a flux is to lower the process temperature, which worsens the penetration of the charge due to heat losses for the melting of calcium oxide.
Point 3. The figures should be better represented, like the Figure 5. In three separate figures, it is hard for readers to compare.
Response 3. Graphs in Fig. 4 and photos in Fig. 5 have been enlarged.
Point 4. The legend in Fig 8 is too small, hard to read.
Response 4. The picture is enlarged
Reviewer 2 Report
Review of the manuscript "Kinetics of the synthesis of aluminum boride by the SHS 2 method" by S.Kh. Aknazarov, A.Zh. Mutushev, Juan Maria Gonzalez-Leal, O.S. Bairakova, O.Yu. Golovchenk, N.Yu. Golovchenko, E.A. Ponomareva.
The paper presents the results of studying the influence of various factors on the kinetics of the process of obtaining aluminum borides by the method of self-propagating high-temperature synthesis.
The topic is interesting, but there are a lot of comments on the work.
1. Quantitative and qualitative values ​​of the obtained results should be added to the abstract.
2. The introduction is a review of only 4 sources, therefore it is perceived as not reasoned.
3. The section materials and methods is not detailed enough. In the results section, a large number of research methods and options are added, which were not even mentioned in section 2.
4. The authors should pay attention to the spelling and style of the English language. For example, fig. 1 - it is better to use the term "scheme" rather than "diagram", line 262 "kids" instead of "samples". Line 113 - "is selected" would be better replaced with "is chosen", "spec" instead of "speck" in table 4, etc.
5. There are abbreviations in the text without explanations. For example, LATR (line 67), MA (245, 246, 260, 261).
6. In the description of the installation (line 65-66) it is not clear which holes are meant. They are not shown in the picture itself.
7. The circuit itself is very primitive. Maybe add a photo of the setup?
8. On line 78 of section 2, the authors refer to the equation presented in the next section. It is not correct.
9. Line 83-85 the authors write that the Gibbs energy has been calculated. Since the calculations themselves are not presented, it is not clear for what temperature the value “-577 kJ/mol” is determined.
10. Fig. 2. The caption of the figure indicates that samples are shown obtained with different ratios of the components indicated in table 1. However, in table 1 there are 9 variants of mixtures, and there are only 5 samples in the figure. In addition, the dimensions of the samples are not clear.
11. Table 2 title should be checked.
12. Everywhere in the text, the authors use the concept of “aluminum size”, obviously we are talking about the size of aluminum powder particles?
13. In table 2, the parameter "Combustion burning rate" and its dimension are not clear. Burning speed should be g/sec.
14. Fig. 3 it is necessary to sign the line numbers in the graph field.
15. Table 3 introduces a new term "Gorenje burning rate, g/s"? What is it?
16. The values ​​of the points on graph 4 do not correspond to the data in table 4. How are the values ​​of the last points on the lines obtained?
17. Table 4. Rows 1 and 7 describe identical experimental conditions, but are the notes different for them?
18. Fig. 5. Thermal analysis results are not readable.
19. Missing table 5.
20. Fig. 6. The text of the signature of the figure and on it duplicate each other. The size of the given particles is not clear.
21. The results of X-ray phase analysis (Fig. 8) raise questions. Where did feldspar come from. There was no silicon in the mixture.
In general, I think that in this form it makes no sense to recommend the article for publication. The material is not systematized, the goals and objectives of the work are not clarified by the authors. As the material is presented, the authors pose and solve more and more new tasks that are not indicated in the tasks of the work. I can advise the authors to use the experiment planning matrix to solve the set extremal problem.
Author Response
Please find enclosed in the document the response to reviewer's comments.

Reviewer 3 Report
The Figures are rather poor, many time there is missing scale, axis label or is it unclear, small and of low resolution, not reader-friendly. Important changes must be done - follow please my yellow notes in the original manuscript file.

Author Response
Response to Reviewer 3:
Point 1. Express it in whole words, not only abbreviation.
Response 1: LATR: Laboratory autotransformer.
Point 2. Why exactly this material - comment it, please.
Response 2: We studied the effect of the amount of heating additive, which was chosen as potassium nitrate, on the rate of the process.
Point 3. What is the meaning? kings
Response 3: It means molten metal. It has been corrected in revised version.
Point 4. Information value of this image is low.Scale bar is missing. What should be here interesting - color change? Some morphological factors? Comment or delete.
Response 4: Pictures have been improved in the revised version.
Point 5. Gorenje
Response 5: It has been corrected in revised version.
Point 6.? (Only a half of some word -?) Spec
Response 6: It has been corrected in revised version.
Point 7. Should be better explained in the context of Tab.1.
Response 7: It has been corrected in revised version.
Point 8. Should be said with full words.
Response 8: It has been corrected in revised version.
Point 9. This is some instrumental output without any try to make it reader-friendly. Axes description, coordinates, units, sample identification - all is improper. Modify it markedle, please, or delete.
Response 9: Pictures have been improved in the revised version.
Point 10. Not very clear sentence.
Response 9: It has been corrected in revised version. The yield from the calculated value was 76.8%.
Point 11. Add scale bar into the photo. Make the XRD pattern better labelled with quartz, feldspar etc (use colours - ?). The signatures are not visible.
Response 11: Pictures have been improved in the revised version.
Point 12. Could be improved.
Response 8: It has been corrected in revised version.
Point 13. Use the subsript for chemical formulas. Superscipts for volumetric units, etc.
Response 13: It has been corrected in revised version.
Round 2
Reviewer 2 Report
1. the abstract remained unchanged. It is necessary to add information about the values ​​of the factors (degree of grinding of the charge, materials, amount of reducing agent, flux and heating additives, degree of charge compaction) that ensure the optimal kinetics of the combustion process.
2. The numbering of references to literary sources in the text of the article has not been updated.
3. The installation description mentions holes at a distance of 5 cm from the thermocouples. For what purpose are they made, how many of these holes are in the lower or upper part of the crucible. I did not find answers to these questions.
4. The authors' answer to remark 9 is not clear. The article indicates the value of the Gibbs energy -577 kJ / mol. In the authors' response "Gibbs energy was calculated depending on the temperature in the range from 298 to 1900 K, the value of ∆G from -548.79 to -273.65 kJ / mol, respectively, indicates the reaction proceeding in a spontaneous mode". Does the process proceed spontaneously at temperatures below 298K? Why hasn't the text been changed?
5. In the answer, the authors write that they have corrected the heading of table 2 (according to remark 11). However, in the new version of the manuscript, it is still not correct.
6. The same applies to remark 12. In the answer, the authors write that they have corrected, but everything in the text is unchanged.
7. Remark 14 was also left without attention. In addition, it can be noted that on the right ordinate of graph 2% is not indicated - did the authors mean atomic or mass?
8. Why is graph 4 changed only in the authors' response file, and in the revised manuscript - the original one, on which there were comments?
9. According to remark 17 no corrections have been made in the text.
10. Remark 19 has not been corrected.
11. Remark 20 has not been corrected.
Author Response
Attached is the response to the reviewer's comments.

Reviewer 3 Report
Some of my advices were accepted, mainly in the text and Tables.
But Figs. 5 and 6 are again inadequate and poorly integrated in the text.
From this standpoint a major revision is necassary.
Author Response

(The authors gave the same response as above.)

Round 3
Reviewer 2 Report
Thanks to the authors for careful revision of the manuscript.
Reviewer 3 Report
My verdict is Accept. The authors deleted some problematic-quality Figures, mainly photos. It is a pity. The best would be to use them, but modify. However, in this form, without these figures, the article is also valuable.